# Simulated trawling: Exhaustive swimming followed by extreme crowding as contributing reasons to variable fillet quality in trawl-caught Atlantic cod (*Gadus morhua*)

**Ragnhild Aven Svalheim**[1]*, **Øyvind Aas-Hansen**[1¤a], **Karsten Heia**[1], **Anders Karlsson-Drangsholt**[2¤b], **Stein Harris Olsen**[1], **Helge Kreutzer Johnsen**[2]

**1** Nofima—The Food Research Institute, Tromsø, Norway, **2** Norwegian College of Fishery Science, Faculty of Biosciences, Fisheries and Economics, University of Tromsø, Tromsø, Norway

¤a Current address: Norwegian Radiation and Nuclear Safety Authority, Section High North, The Fram Centre, Tromsø, Norway
¤b Current address: The Research Council of Norway, Oslo, Norway
* ragnhild.svalheim@nofima.no

## Abstract

Trawl-caught Atlantic cod (*Gadus morhua*) often yield highly variable fillet quality that may be related to capture stress. To investigate mechanisms involved in causing variable quality, commercial-sized (3.5±0.9 kg) Atlantic cod were swum to exhaustion in a large swim tunnel and subsequently exposed to extreme crowding (736±50 kg m$^{-3}$) for 0, 1 or 3 hours in an experimental cod-end. The fish were then recuperated for 0, 3 or 6 hours in a net pen prior to slaughter to assess the possibility to reverse the reduced fillet quality. We found that exhaustive swimming and crowding were associated with increased metabolic stress, as indicated by increased plasma cortisol, blood lactate and blood haematocrit levels, accompanied by reduced quality of the fillets due to increased visual redness and lower initial muscle pH. The observed negative effects of exhaustive swimming and crowding were only to a small degree reversed within 6 hours of recuperation. The results from this study suggest that exhaustive swimming followed by extreme crowding can reduce fillet quality, as measured by fillet redness and muscle pH, and contribute to the variable fillet quality seen in trawl-caught Atlantic cod. Recuperation for more than six hours may be required to reverse these effects.

## Introduction

Fish captured in a trawl encounter a number of strenuous and stressful events such as forced swimming, crowding, confinement, crushing and barotrauma [1]. A trawl is a cone shaped net made from two, four or more panels, which is towed by one or two boats [2]. Common towing speeds for cod fishing vary from 2–5 knots (1 to 2.5 m s$^{-1}$). Trawling triggers a complex sequence of behavioural responses by the captured fish, which include trying to avoid the

**Data Availability Statement:** All relevant data are within the paper and its Supporting Information files.

**Funding:** This study was part of the Centre of Research-based Innovation in Sustainable fish capture and Processing technology (CRISP) project funded by the Norwegian Research Council (https://www.forskningsradet.no/en/), Grant No. 203477. The funders had no role in study design, data collection and analysis, decision to publish, or preparation of the manuscript.

**Competing interests:** The authors have declared that no competing interests exist.

approaching trawl and fleeing from the net opening by swimming at the same speed and direction as the trawl with a 'kick and glide' style [3]. The towing speed during trawling exceeds the sustainable swimming speed of cod [4], which indicates that the fish will be exhausted once they enter the cod-end. When the number of fish increase in the cod-end, the animals will be compressed resulting in a stressful and extreme crowding situation.

Short-term stress causes an elevation of circulating catecholamines and corticosteroids (e.g. cortisol), which in turn will alter metabolism, osmotic balance and increase heart and ventilation rates. An ultimate function of the short-term stress response is mobilization of stored fuels for the physiological reactions known as "fight or flight". Physiological measurements of trawl-captured cod show fish in near homeostatic crisis with a variable degree of muscle redness. This pre-mortem stress can strongly influence the quality of the final fish product [5] as it is associated with textural changes of the fish meat by altering the rate and extent of pH decline, causing a more rapid onset of rigor mortis [6, 7]. Furthermore, pre-mortem stress is associated with a change in muscle colour, which is considered an aesthetic quality defect in white fish [8] causing a discolouration of the fish product and resulting in economic loss for the producer [9]. Therefore, finding ways to reduce or reverse detrimental effects of capture stress may be of economic interest for both fishermen and producers.

During commercial trawling, it is challenging to separate the various parameters that can induce stress and reduce fillet quality. Variable size and length of the hauls is of great importance to both fillet colour and survival of the catch in commercial trawling [10], but it is not possible to distinguish between the different stressors affecting fish in a trawl situation (e.g. exhaustive swimming, crowding or barotrauma) onboard a trawler. Investigating trawl-related stress in an experimental setting may give a better understanding of how the amount of residual blood in fillets is influenced by different pre-mortem stressors. Previously, we have shown that neither the poor physiological state or negative fillet quality features of trawled cod could be reproduced by exhaustive swimming alone and argue that variable fillet quality more likely is the result of several factors operating during the trawling process [11, 12]. In addition, a study performed onboard commercial trawlers has shown that it is possible to improve the fillet colour of cod by keeping them alive in holding tanks for a few hours prior to slaughter [10].

In the current study, our aim was to experimentally simulate the exhaustive swimming followed by extreme crowding experienced by fish during commercial trawl capture and investigate how this affects some key metabolic stress parameters and subsequent fillet quality traits (amount of blood in fillets and muscle pH) in wild-caught Atlantic cod. A second aim of the study was to investigate if post-stress recuperation for 0, 3 or 6 hours could reverse potential negative effects on stress and selected fillet quality traits.

## Materials and methods

### Animals and husbandry

Wild Atlantic cod were captured by Danish seine in mid May 2014 outside the coast of Finnmark, Norway. The fish were kept alive on board in tanks supplied with running seawater for 12 hrs and delivered to a live fish storage facility in Nordvågen, Norway, for recuperation for three weeks without feeding. From there, the fish were transported approximately 300 km in a wellboat to Tromsø Aquaculture Research Station at Ringvassøy, Norway. At the research station, the fish were held in two outdoor tanks (4 m diameter, 10 m$^3$) supplied with filtered seawater at natural water temperature and day-length (69°N), until the start of the experiment in February 2015. The fish were fed three times a week, using a mixture of capelin (*Mallotus villosus*) and commercial feed (Skretting Amber 5 mm, Skretting ASA, Norway) during the holding time and between each trial. Feeding stopped 48 hours before transfer of fish into an outdoor

swimming tunnel (1400 L swim chamber, maximum speed 1.2 ms$^{-1}$, we have previously described tunnel in detail Karlsson-Drangsholt, Svalheim (11)). Biological data ofthe fish were obtained on each sampling day (body mass 3.5 ± 0.9 kg, body length 75 ± 7 cm, mean ± SD) (mean and SE for each run in run Table 1). There were no differences in gender distribution (N = 114 females and N = 95 males).

## Experimental set-up and animal welfare

A total of 211 fish were used in the experiment. The experiment was conducted in three replicates (trials) over 26 days (see and Fig 1 and Table 1 for details). Each trial consisted of two runs with 28 fish in each; one run with 1 hour crowding time following swimming and one run with 3 hours crowding time following swimming (Fig 1; Table 1). For both runs, 7 fish were sampled as swum control immediately after swimming (described below). Following crowding, 7 fish were immediately sampled (0 hours recuperation), whereas 7 fish were recuperated for 3 hours and 7 fish were recuperated for 6 hours, before being sampled. (Fig 1; Table 1). Three crowding durations of 1, 3 and 5 hours were selected in the original set-up to represent short, medium and long trawl hauls based reports from commercial trawl hauls [10]. In the first trial, however, mortality in the 5 hour crowding group was 91% (Table 1) and this group was therefore omitted in subsequent trials on grounds of animal welfare. No fish were removed from the experimental cod-end during crowding as this would reduce the crowding pressure and induce additional handling stress, thereby interfering with the experimental conditions. However, after the crowding, any individuals that exhibited signs of significant injury or distress were removed immediately and euthanized with a blow to the head. The experimental design was reviewed and approved by the Norwegian Animal Research Authority (permission FOTS 6910) and all research staff were licensed to work with research animals.

Diagram showing the experimental design of the present study. The experiment was performed as three separate trials, with each trial consisting of two runs on separate days (see Table 1 for details). In each trial, 7 rested control fish were sampled from the holding tanks and 28 fish were transferred to the swim tunnel for acclimation and subsequent swimming. In the first run of each trial, 28 fish were swum, after which 7 swum control fish were sampled and the 21 remaining fish were crowded in the cod-end for one hour. In the second run of each trial, all fish were swum, after which 7 swum control fish were sampled and the 21 remaining fish were crowded in the cod-end for three hours. Following crowding in both runs, 7 fish were immediately sampled (0 hrs recuperation), 7 fish were recuperated for 3 hours and then sampled (3 hrs recuperation), and 7 fish were recuperated for 6 hours and then sampled (6 hrs recuperation).

**Rested control fish.** Two days before each trial, 7 fish were randomly dip-netted from the two holding tanks. 3 fish were taken from one tank and 4 from the other. These fish were used to establish baseline levels for measured parameters for rested, unstressed fish (rested control). The fish were captured and sampled within 1 minute.

**Exhaustive swimming trial.** Before each run, 28 fish were transferred to the swim tunnel and acclimated for 36 hours at a water speed of 0.15 m s$^{-1}$. The fish density in the tunnel was on average 54 kg m$^{-3}$. The swimming trial commenced with increasing water velocity from 0.15 to 1.2 m s$^{-1}$ in 1200 steps in 20 minutes (1 step s$^{-1}$; 0,000875 m s$^{-1}$ step$^{-1}$), yielding a near continuous and linear speed increase. Fish were monitored via two video cameras placed in the front and side of the tunnel. As fish ceased swimming and rested on the grid of the 1$^{st}$first hatch at the back of the tunnel (Fig 2), they were pinched in the tail with use of fingers to see if they would continue swimming. Non-responsive fish were considered exhausted [12] and subsequently released into the retention chamber, where water flow (approx. 0.8 m s$^{-1}$ at max.

**Table 1. Overview of all trials and runs.** Overview of trials and runs, treatment (Rested control (RC), Swum control (SC) 1 hr crowding (1hC), 3 hrs crowding (3hC) and 5 hrs crowding (5hC)), date, number of fish in each run (N), biological data (gender, weight, length condition factor (CF), gonadosomatic index (GSI), hepatosomatic index (HIS), mean ± SE), fish density during crowding in the cod-ed (FD), mortality (Mort. %) per run, air temperature and water temperature for each day of the trial period. OM are runs omitted from experiment because of high mortality.

| Trial no. | Run | Treatment | Recupe-ration time (hrs) | Date | ♂ | ♀ | N | Weight | Length | CF | GSI | HSI | FD (kg m-3) | Mort. (%) | Air temp. °C | Water temp °C |
|---|---|---|---|---|---|---|---|---|---|---|---|---|---|---|---|---|
| 1 | 1 | RC | Na | 01.02.2015 | 2 | 5 | 7 | 3771 ± 503 | 75 ± 3 | 0.85 ± 0.07 | 6.23 ± 3.47 | 4.80 ± 0,412 | Na | 0 | −5 ± 0.4 | 3.4 |
|  |  | SC | 0 | 03.02.2015 | 3 | 4 | 7 | 3939 ± 233 | 77 ± 2 | 0.88 ± 0.15 | 7.64 ± 2.39 | 4.38 ± 0,525 | Na | 0 |  |  |
|  |  | 1hC | 0 | 03.02.2015 | 3 | 4 | 7 | 4211 ± 448 | 78 ± 4 | 0.89 ± 0.19 | 9.57 ± 2.32 | 4.76 ± 0,612 | 803 | 0 | −7.8 ± 0.4 | 3.4 |
|  |  | 1hC | 3 | 03.02.2015 | 2 | 5 | 7 | 4171 ± 425 | 81 ± 2 | 0.78 ± 0.12 | 4.87 ± 1.59 | 4.41 ± 0,669 |  |  |  |  |
|  |  | 1hC | 6 | 03.02.2015 | 6 | 1 | 7 | 3343 ± 317 | 73 ± 4 | 0.92 ± 0.38 | 5.69 ± 2.01 | 4.48 ± 0,665 |  |  |  |  |
|  | 2 | SC | 0 | 10.02.2015 | 2 | 5 | 7 | 3397 ± 461 | 75 ± 3 | 0.78 ± 0.08 | 3.12 ± 1.13 | 3.98 ± 0,34 | Na | 0 |  | 3.5 |
|  |  | 3hC | 0 | 10.02.2015 | 2 | 5 | 7 | 3603 ± 304 | 77 ± 2 | 0.78 ± 0.14 | 7.03 ± 2.57 | 3.93 ± 0,412 | 802 | 48 | −0.9 ± 1.1 |  |
|  |  | 3hC | 3 | 10.02.2015 | 5 | 2 | 7 | 3934 ± 94 | 80 ± 2 | 0.78 ± 0.13 | 6.19 ± 1.69 | 3.83 ± 0,386 |  |  |  |  |
|  |  | 3hC | 6 | 10.02.2015 | 3 | 5 | 8 | 3645 ± 211 | 77 ± 2 | 0.79 ± 0.13 | 6.08 ± 2.03 | 4.34 ± 0,336 |  |  |  |  |
| 2 | 3 | RC | Na | 08.02.2015 | 1 | 6 | 7 | 3626 ± 344 | 76 ± 2 | 0.81 ± 0.11 | 3.48 ± 1.64 | 4.34 ± 0,491 | Na | 0 | −6 ± 0.4 | 3.5 |
|  |  | SC | 0 | 12.02.2015 | 2 | 5 | 7 | 3293 ± 336 | 75 ± 2 | 0.77 ± 0.15 | 4.79 ± 2.12 | 4.69 ± 0,491 | Na | 0 |  |  |
|  |  | 1hC | 0 | 12.02.2015 | 2 | 5 | 7 | 2683 ± 227 | 68 ± 2 | 0.83 ± 0.12 | 5.79 ± 2.64 | 3.56 ± 0,506 | 672 | 0 | −4.9 ± 0.8 | 3.5 |
|  |  | 1hC | 3 | 12.02.2015 | 4 | 3 | 7 | 3423 ± 322 | 75 ± 2 | 0.79 ± 0.11 | 4.73 ± 1.84 | 3.40 ± 0,544 |  |  |  |  |
|  |  | 1hC | 6 | 12.02.2015 | 4 | 3 | 7 | 3706 ± 336 | 76 ± 3 | 0.84 ± 0.06 | 1.74 ± 0.83 | 5.28 ± 0,435 |  |  |  |  |
|  | 4 | SC | 0 | 17.02.2015 | 2 | 5 | 7 | 3418 ± 267 | 73 ± 2 | 0.87 ± 0.18 | 5.75 ± 2.25 | 4.72 ± 0,809 | Na | 0 |  |  |
|  |  | 3hC | 0 | 17.02.2015 | 4 | 3 | 7 | 3776 ± 369 | 76 ± 4 | 0.88 ± 0.17 | 7.37 ± 2.69 | 5.30 ± 0,556 | 706 | 5 | 1.3 ± 0.54 | 3.6 |
|  |  | 3hC | 3 | 17.02.2015 | 2 | 5 | 7 | 3304 ± 417 | 74 ± 4 | 0.79 ± 0.17 | 8.49 ± 3.13 | 4.22 ± 0,658 |  |  |  |  |
|  |  | 3hC | 6 | 17.02.2015 | 4 | 3 | 7 | 3222 ± 172 | 71 ± 2 | 0.91 ± 0.11 | 9.29 ± 3.32 | 4.25 ± 0,461 |  |  |  |  |
| 3 | 5 | RC | Na | 22.02.2015 | 4 | 3 | 7 | 3034 ± 296 | 72 ± 3 | 0.82 ± 0.12 | 3.27 ± 1.33 | 4.07 ± 0,484 | Na | 0 | −1.3 ± 2.27 | 3.6 |
|  |  | SC | 0 | 24.02.2015 | 3 | 4 | 7 | 3364 ± 339 | 72 ± 2 | 0.90 ± 0.11 | 5.11 ± 1.78 | 4.51 ± 0,321 | Na | 0 |  |  |
|  |  | 1hC | 0 | 24.02.2015 | 4 | 3 | 7 | 3567 ± 204 | 74 ± 2 | 0.87 ± 0.07 | 4.34 ± 1.63 | 4.65 ± 0,465 | 733 | 0 | 0.9 ± 1.0 | 3.6 |
|  |  | 1hC | 3 | 24.02.2015 | 3 | 4 | 7 | 3690 ± 178 | 75 ± 1 | 0.86 ± 0.10 | 5.48 ± 4.40 | 4.78 ± 0,249 |  |  |  |  |
|  |  | 1hC | 6 | 24.02.2015 | 2 | 5 | 7 | 3446 ± 309 | 73 ± 2 | 0.86 ± 0.10 | 3.62 ± 1.37 | 4.78 ± 0,514 |  |  |  |  |
|  | 6 | SC | 0 | 26.02.2015 | 5 | 2 | 7 | 2608 ± 256 | 69 ± 3 | 0.81 ± 0.17 | 3.26 ± 1.24 | 3.44 ± 0,529 | Na | 0 |  | 3.5 |
|  |  | 3hC | 0 | 26.02.2015 | 5 | 2 | 7 | 3808 ± 230 | 76 ± 2 | 0.86 ± 0.09 | 5.78 ± 193 | 4.52 ± 0,552 | 702 | 0 | 0.1 ± 1.1 |  |
|  |  | 3hC | 3 | 26.02.2015 | 4 | 3 | 7 | 2836 ± 348 | 72 ± 3 | 0.74 ± 0.11 | 0.43 ± 0.08 | 4.56 ± 0,964 |  |  |  |  |
|  |  | 3hC | 6 | 26.02.2015 | 4 | 3 | 7 | 3604 ± 412 | 73 ± 2 | 0.92 ± 0.11 | 3.04 ± 1.22 | 5.71 ± 0,51 |  |  |  |  |
| OM | OM | SC | 0 | 05.02.2015 | 2 | 5 | 7 | 3331 ± 435 | 71 ± 3 | 0.92 ± 0.06 | 7.94 ± 2.48 | 4.76 ± 0.52 |  |  | −8.7 ± 3.3 | 3.4 |
|  |  | 5hC | 0 | 05.02.2015 | 3 | 4 | 7 | 3451 ± 291 | 72 ± 2 | 0.94 ± 0.07 | 5.39 ± 0.60 | 5.07 ± 0.44 | 686 | 91 |  |  |
|  |  | 5hC | 3 | 05.02.2015 | 9 | 5 | 14 | 3128 ± 282 | 73 ± 1 | 0.81 ± 0.06 | 6.69 ± 3.18 | 4.44 ± 0.76 |  |  |  |  |

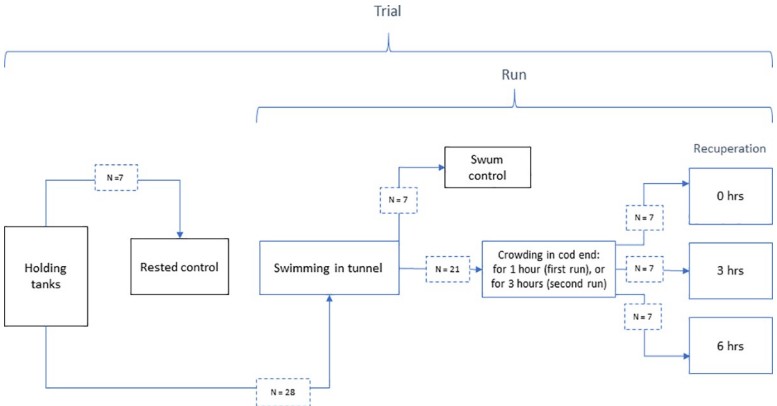

**Fig 1. Experimental design.**

speed) kept them on the grid of the 2<sup>nd</sup>second hatch (Fig 2). For all runs, it took maximum 15 minutes from the first to the last fish to become exhausted and thus enter the retention chamber. A pre-randomized list based on the order of exhaustion made separately for each run was used to select fish for the swum control. As a fish destined for the swum control group became exhausted and entered the retention chamber, it was immediately netted out and sampled.

**Crowding in the experimental cod-end.** Following the exhaustive swimming and sampling of swum control fish, the remaining 21 fish were released directly from the retention chamber into an experimental cod-end (Fig 2). The cod-end was constructed as a four-panel cylindrical bag (length 200 cm height 58 cm with tension) using the same material as in a commercial cod-end (8 cm diamond cod-end mesh, 0.3 cm twine). The cod-end could be opened via a joint at the top (Fig 2). A rope was placed at a fixed position to close the cod-end and tightened to ensure that the fish were crowded (Fig 2). When the cod-end was closed it was roughly sphere-shaped with a diameter of about 58 cm (S1 Fig) yielding a volume of about 70 L. In each run, fish density was estimated based on the average weight of total individuals in the cod-end (Table 1). The fish were visually inspected every 15 minutes. Oxygen inside the cod-end was continuously monitored using an YSI ProODO handheld dissolved oxygen

**Fig 2. Schematic overview of the swim tunnel/trawl simulator.** Graphic illustration of the swim tunnel and fish chamber, retention chamber and the experimental cod-end.

meter with a ProODO Optical probe (Yellow Spring Instruments, Ohio, USA). The fish were crowded for either 1 hour (first run in each trial) or 3 hours (second run in each trial). When the crowding time was over the fish were taken out of the experimental cod-end and sampled immediately (0 hours recuperation) or recuperated for 3 or 6 hours in a random fashion.

### Recuperation

The 3 and 6 hours recuperation groups were kept in 1×1×1 m lid-covered steel mesh (mesh size 4×4 cm) cages placed in an 11 m diameter fiberglass tank supplied with running seawater at natural water temperature (3.5 ± 0.8˚C Feb-March) to ensure flow-through of oxygen-saturated water. The fish were inspected visually every 15 minutes.

### Sampling procedure

All fish were euthanized by a blow to the head and blood was collected from the caudal vessels within 1 minute, using 4 ml heparinized vacutainers with 4×0.9 mm needles (BD Diagnostics, Franklin Lakes, NJ, USA). Measurements of pH were then obtained by inserting a Hamilton double pore glass electrode (WTW330/set-1 pH meter,Wissenscaftliche-Technische Werkstätten, Weilheim, Germany. Electrode: Hamilton Bonaduz AG, Bonaduz, Switzerland) via an incision (1 cm×2 cm) in the epaxial part of the white muscle tissue, rostrally to the dorsal fin on the left side of the fish. During the post-mortem pH measurements, a new incision was subsequently made 1 cm caudal to the previous incision for each measurement. pH was measured immediately after euthanasia, then there was a 20 hour period without measurements followed by measurements approximately every 8–15 hour. The instrument was calibrated frequently using pH 4.01 and 7.00 buffers at 2˚C, and the electrode was cleaned with demineralized water between each measurement.

Concentrations of lactate and glucose were measured in samples of whole blood, using the hand-held meters Lactate Scout+ (SensLab GmbH, Germany) and FreeStyle Lite (Abbott Diabetes Care, Inc., Alameda, CA), respectively. Haematocrit measurements were performed with a microhaematocrit capillary tube reader (Critocaps; Oxford Lab, Baxter, Deerfield, IL). The remaining blood was then centrifuged at $2700 \times g$ for 5 minutes at 4˚C, and plasma was transferred to cryo tubes, frozen in liquid nitrogen and stored at −80˚C for later analysis of plasma cortisol. Immediately after blood collection and peri-mortem pH-measurements, all fish were exsanguinated by cutting the *Bulbus arteriosus* and *Vena cardinalis communis* on both sides. The fish were then bled for 30 minutes in a tank supplied with running seawater. Afterwards, weight (g), length (cm) and gender of each fish were registered. The liver and gonads were then taken out and weighed (g) to determine hepatosomatic (HSI) and gonadosomatic indices (GSI) by tissue weight x 100/total weight. The fish were then gutted, covered with plastic film and placed on ice in standard plastic fish boxes and stored at 4˚C.

### Fillet redness

After approximately 72 hours of storage all fish were filleted by trained personnel. The fillets were not de-skinned, but the black lining of the peritoneum was removed. Each fillet was evaluated by a sensory panel of three trained and experienced persons. To avoid expectation bias, the sensory panel was unaware of which group of fish they were evaluating. The sensory panel evaluated each fillet by general redness, blood filled vein and blood spots. The fillets were given a final score from 1 to 4, where 1 was a white fillet and 4 was a clearly red fillet.

## Imaging VIS/NIR spectroscopy

After filleting, the muscle haemoglobin was evaluated by hyperspectral imaging of the fillets in diffuse reflectance mode. Imaging was performed with a push-broom hyperspectral camera with a spectral range of 430–1000 nm and spatial resolution of 0.5 mm across-track by 1.0 mm along track (Norsk Elektro Optikk, model VNIR-640, Skedsmokorset, Norway). The camera was fitted with a lens focused at 1000 mm, and mounted 1020 mm above a conveyor belt. Light penetrated the fillet to a depth of 2 mm and by characterizing the diffuse reflectance, those spectra were transformed into absorbance spectra. Following the procedure outlined by Skjelvareid et al. [13], the haemoglobin concentration in milligram haemoglobin per gram muscle was estimated on pixel level for each fillet.

## Cortisol analysis

Plasma concentrations of cortisol were analysed by use of radioimmunoassay (RIA), in accordance with previously described methods [14, 15]. In short, cortisol was extracted from 300 μL plasma with 4 mL diethyl ether under shaking for four minutes. The aqueous phase was frozen in liquid nitrogen and the organic phase was decanted to tubes and evaporated in a water bath at 45˚C for ca 20 minutes and reconstituted by addition of 900 μL assay buffer before analysed by RIA. The antibody used was obtained from New Zealand white (NZW) rabbits and the detection limit for the assay was 0.6 ng mL$^{-1}$ [14].

## Statistical analysis and data management

The data was analysed with the statistical software R, version 3.4.0 [16]. The relationships between response variables (plasma cortisol (ng L$^{-1}$), lactate (mmol L$^{-1}$), glucose (mmol L$^{-1}$), pH, fillet redness, muscle pH) and corresponding potential explanatory variables (as factor; groups: crowding 1 or 3 hours, recuperated 0, 3 o 6 hours, rested control and swum control), sex (as factor), plasma cortisol, blood glucose, blood lactate, muscle haemoglobin (mg g$^{-1}$), hepatosomatic index (HSI), gonadosomatic index (GSI) and Fulton's condition factor (100 g cm$^{-3}$), were investigated using Generalised Linear Modelling (GLM) [17]. Muscle pH was modelled with time post-mortem and groups: crowding 1 or 3 hours, recuperated 0, 3 or 6 hours, rested control and swum control and curvature were checked by testing with different polynomials and interactions to determine significant differences between slopes. Note that some variables are both response and explanatory, depending on which response is under investigation. Before proceeding with the GLM analysis, the data was checked and prepared for modelling following procedures previously described [18].

Most of the response variables had only positive values and were therefore best modelled using Gamma distribution, which accounts for skewed distribution of model errors and prevents negative predictions. In those cases where distribution was normal and there was no risk of predicting negative values, data was modelled using Gaussian (Normal) error distribution. In the case for sensory evaluation of redness, data were strictly bound between 1 and 4 and therefore fitted to a quasi-binomial distribution to make sure that predicted values also fall within this range. Link function (identity, log, inverse or logit) was chosen based on which link gave the best fit to data in terms of lowest Akaike information criterion (AIC) and by visual evaluation of the graphics. All model details are available in S1 Model.

## Results

Fish density in the cod-end varied between trials from 672 to 803 kg m$^{-3}$(Table 1) and the oxygen saturation of the water inside the cod-end always remained above 95% at any time and

position within the cod-end. There were no mortalities during the swim-trial (i.e. swim tunnel and retention chamber) or following crowding for one hour, but for the group crowded for 3 hours 18% of the fish were considered dead or moribund. The first run with 3 hours crowding had 48% mortality, whereas the last two runs had 5 and 0% mortality, respectively (Table 1).

The plasma level of cortisol was clearly affected by swimming, crowding and recuperation (p < 0.001), but was also correlated with GSI (p <0.001) (S2 Fig of Fig 1). The fish that were only swum (and not crowded) experienced a slight increase in plasma cortisol compared to the resting control. The highest levels of cortisol were found immediately after crowding for 3 hours and after 3 hours recuperation for the 1 hour crowding group. After 6 hours of recuperation, the cortisol levels were still elevated (Fig 3A).

Blood glucose was affected by crowding and recuperation (p<0.001) and was positively correlated with HSI (p < 0.001) (S2 Fig of Fig 3). Blood glucose was higher after crowding for 1 and 3 hours compared to both resting and swum controls and remained elevated throughout the recuperation period (Fig 3B).

Blood lactate was clearly affected by swimming (p<0.001) and duration of crowding (p<0.001) (Fig 3C). Fish crowded for 1 hour had significantly higher lactate levels compared to resting and swum control (p<0.001) and the levels remained elevated throughout the recuperation period. The animals crowded for 3 hours showed a 2-fold increase in lactate levels compared to 1 hour (p<0.001). The lactate stayed elevated throughout the recuperation period. Blood lactate levels were also negatively correlated to muscle pH (p<0.001) (S2 Fig of Fig 3), this correlation was strongest for the 3 hours crowding group.

Fillet redness was affected by swimming, crowding and recuperation and was positively correlated with muscle haemoglobin levels (S2 Fig of Fig 4). There were no major differences between fillets of fish crowded for 1 hour versus those crowded for 3 hours. After 6 hours of recuperation, the level of redness was still higher than for resting and swum control, but lower than after 0 and 3 hours of recuperation (Fig 4A). In the GLM ran without haemoglobin as explanatory variable, swimming, crowding and recuperation remained significant explanatory variables (p<0.001). In addition, a positive correlation between cortisol level and redness was found (p = 0.043) (S2 Fig of Fig 5).

Crowding and recuperation affected muscle haemoglobin (p = 0.007), but only for the fish crowded for 3 hours without recuperation (Fig 4B). When modelled together with haematocrit, this effect disappeared and only haematocrit remained a significant explanatory variable (p = 0.02) (S2 Fig of Fig 6). Because it can be argued that haemoglobin and haematocrit are dependant, a second GLM without haematocrit was run. In the second run, a positive correlation between cortisol level and muscle haemoglobin was found (p = 0.012). Also, the swimming, crowding and recuperation was significant when modelled together with cortisol (p = 0.008) (S2 Fig of Fig 7).

Swimming, crowding and recuperation had a transient effect on haematocrit (p < 0.001), which was influenced by both crowding and recuperation time and was positively correlated with plasma cortisol levels (p = 0.038) (S2 Fig of Fig 8). The haematocrit response was highest immediately after 1 and 3 hours of crowding and decreased to control levels after 3 hours (Fig 4B).

Muscle pH was affected by swimming, crowding and recuperation (Fig 4D). The peri-mortem pH was lowest in un-recuperated crowded fish, but there were no differences between groups crowded for 1 and 3 hours. However, the fish crowded for 1 hour recovered faster than fish crowded for 3 hours.

## Discussion

There is growing interest in the fishing industry to improve the quality of fish caught by commercial trawlers. The problem is that large catches and lengthy hauls often result in muscle

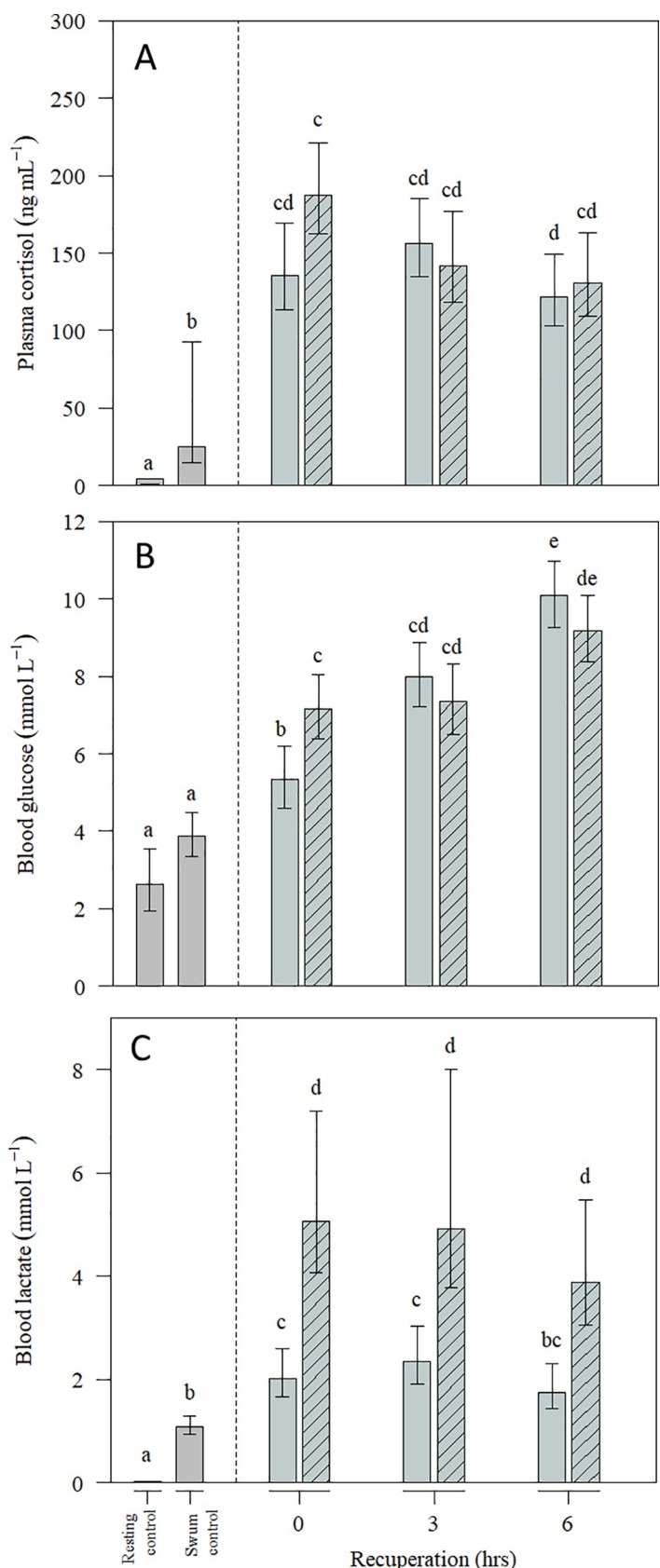

**Fig 3. Physiological stress response to crowding and recuperation.** Plasma cortisol (A), blood glucose (B) and blood lactate (C) in Atlantic cod during recuperation following exhaustive exercise and severe crowding for 1 hour (open bars) or 3 hours (dashed bars). Fish in the rested control group were sampled from the holding tank and the fish in the swum control group were sampled immediately after individual exhaustion. Data are presented as estimated means, and errors indicate 95% confidence intervals from the three runs (N = 21 per group and N = 45 for swum control) fitted from GLM. Overlapping confidence intervals suggests means are not significantly different from each other. See S1 Model.

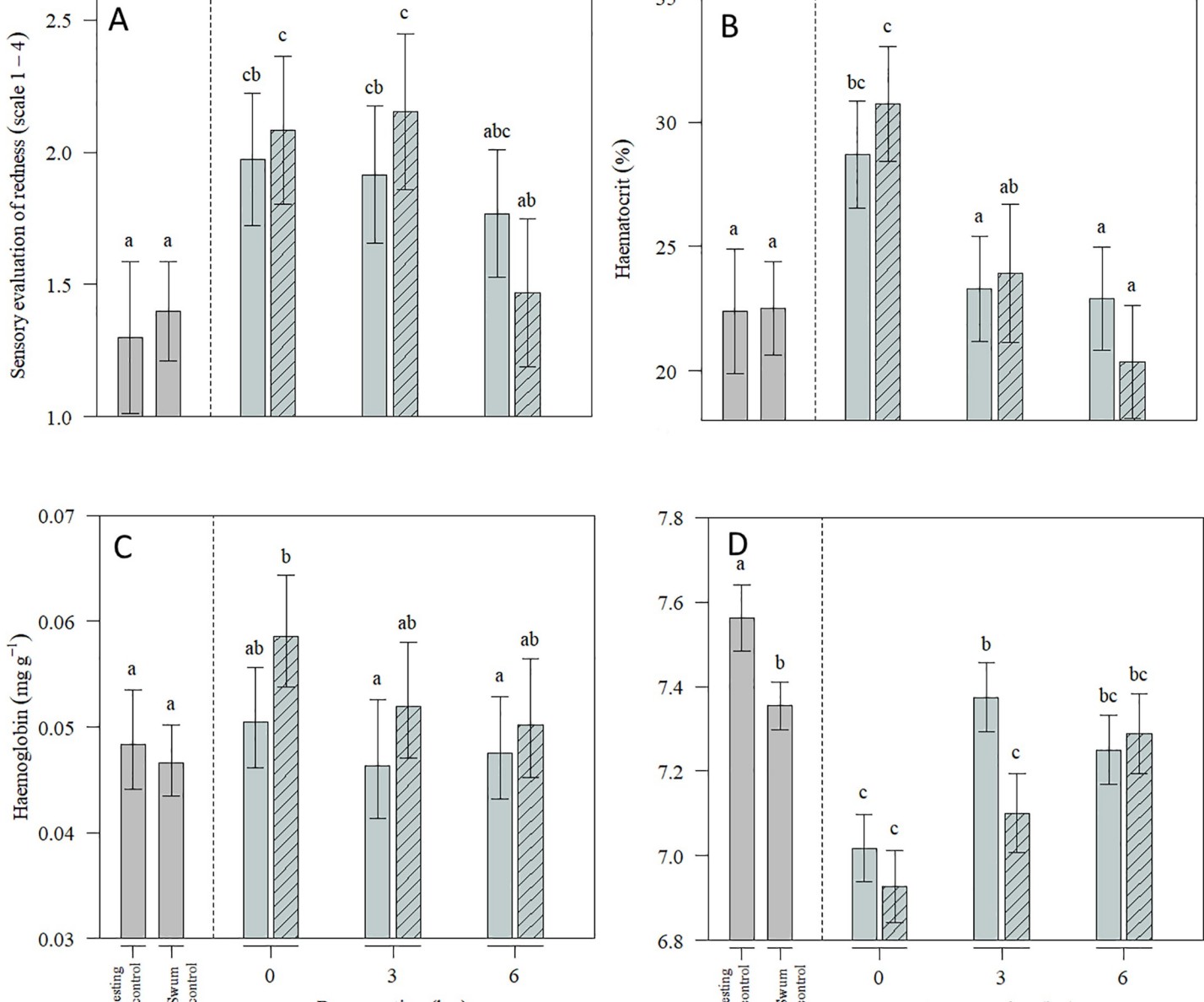

**Fig 4. Redness, haematocrit and muscle haemoglobin.** Sensory evaluation of redness (A), haemotocrit (B) and muscle haemoglobin concentration (mg haemoglobin per gram muscle) in the surface area of fillets, measured by spectroscopy (C), and muscle pH (D)in Atlantic cod during recuperation following exhaustive exercise and severe crowding for 1 hour (open bars) or 3 hours (dashed bars). Fish in the rested control group were sampled from the holding tank and the fish in the swum control group were sampled immediately after individual exhaustion. Data are presented as estimated means, and errors indicate 95% confidence intervals from the three runs (N = 21 per group and N = 45 for swum control) fitted from GLM. Overlapping confidence intervals suggests means are not significantly different from each other. See S1 Model.

segment gaping and a reddish coloration of the fillet. Both are considered quality defects that may lead to downgrading of the fish and financial loss for the producer [19, 20]. One way to circumvent this problem may be to temporarily store the fish live in tanks supplied with running seawater to let the fish recover from the capture process. This procedure has been used successfully to reduce the amount of residual blood in the fillets of Atlantic cod caught by trawl [10].

We found that exhaustive swimming followed by crowding caused a severe metabolic stress response, as demonstrated by high plasma cortisol levels and elevated blood lactate and glucose levels. The metabolic stress was accompanied by a reduction in muscle pH and increased fillet redness, similar to that reported for cod caught by trawl [10, 21]. The direct cause of the stress induced by crowding is not clear, but a gradual build-up of blood lactate, which correlated with the duration of the crowding, is an indication of oxygen deficiency and prolonged anaerobic metabolism during the period of confinement. Our initial expectation was that there would be less oxygen available inside the cod-end during crowding. The hypoxic conditions could in turn reduce the oxygen uptake of the fish. However, in the current experiment water oxygen saturation always remained above 95% at any position inside the experimental cod-end. It seems more likely, therefore, that the cod in our experiment may have experienced hypoxia as a consequence of impaired opercular movement and thus insufficient ventilation due to the very high fish density inside the cod-end.

In the present experiment, post-exercise crowding for 1 and 3 hours, were associated with 0 and 18% mortality after 6 hours of recovery, respectively (Table 1). This suggests that the majority of Atlantic cod can handle extreme crowding (about 700 kg m$^{-3}$) for 3 hours. However, we did find a mortality of 48% in the first run of fish crowded for 3 hours (Table 1). This trial had higher fish density (*i.e.* about 800 kg m$^{-3}$) than the last two trials. The density was however not higher than the first trial with 1 hour crowding. This indicates that crowding time is particularly important when the fish density is high. A study from commercial trawlers found that hauls longer than 5 hours led to up to 27% mortality [10]. This is in contrast to the initial trial in our experiment where confinement in the cod-end for 5 hours resulted in over 91% mortality. We speculate that the discrepancy between our experiment and the observations from commercial trawls, may be due to the gradual filling of the trawl under natural conditions, in which case the fish would not experience extreme crowding until the cod-end is filled up to some degree. For example, another large-scale trawl study found a less severe cortisol response (~ 60 ng mL$^{-1}$) in cod after hauls lasting 15–55 minutes [6], compared to the fish in our study that were confined in the experimental cod-end for 1 hour (~ 200 ng mL$^{-1}$).

During hypoxia, the metabolic fuel preference is thought to shift from mainly lipids and proteins to carbohydrates [22]. We found a marked elevation in blood glucose after crowding, which continued to increase throughout the recuperation period. This is most likely due to catecholamine and cortisol-mediated stimulation of glycogenolysis and gluconeogenesis, respectively, which is not met by a comparable increase in glucose utilisation [23, 24]. We also found that fillet redness in terms of overall colour and residual blood in veins, increased as a response to crowding, and that this correlated with elevated plasma cortisol levels and muscle haemoglobin. This suggests that the sensory evaluation of redness is a valid method for assessing amount of blood in cod fillets. In addition, the haemoglobin measurement was positively correlated with haematocrit, indicating that the method is indeed measuring amount of blood in the fillets [25]. In Atlantic cod, hypoxic conditions are reported to increase resistance of vessels supplying the stomach, intestines and other digestive organs, while somatic circulation is dilated [26], thereby redistributing blood flow to the muscle. Furthermore, in rainbow trout 80% of cardiac output is found to be routed to the white muscle of during recovery from strenuous exercise [27]. Thus, it seems that the increase in haematocrit together with a presumed

increased blood perfusion of the white muscle during recovery, may be the most important factors causing increased redness of the fillet, during recovery in this experiment. Other studies, on the other hand, have found that stress does not necessarily cause an increase in fillet redness [11, 12, 28, 29], and at least one study have found that apparently physiologically unstressed fish still can yield a blood-filled fillet after crowding treatment [30]. One can therefore speculate whether it is the stress itself that causes the fillet redness, or if the method of inducing stress might be of importance to amount of blood in fillets. In the studies where no fillet redness was observed the fish were at all times allowed to swim, and stress was induced by either chasing [29], water reduction [28] or forced swimming [11, 12], whereas this study and other studies used stressors such as crowding, netting and/or air exposure, which heavily restricts body movement [10, 30, 31]. A possible explanation for the difference in residual blood content of white muscle between these studies, may therefore be that accumulation of residual blood in the crowded fish is caused by insufficient emptying/return of venous blood from the segmental veins, due to impaired movement of the swimming muscles. This is because the venous blood from the swimming muscles of fish is passed to the central veins via the segmental veins and depends on the alternating movement of the lateral muscles, which squeezes the blood out of the segmental veins and into the central veins when the ipsilateral muscles contract [32]. The segmental veins, in turn, are guarded by ostial valves, which prevents backflow of blood when the ipsilateral muscles relax. Hence, return of venous blood from the swimming muscles is dependent on the continuous, alternating sideways muscular movement of the body of the fish. This mechanism is similar to the muscular pump of the lower limb in humans [33].

## Conclusion

In the present experiment, exhaustive swimming followed by extreme crowding for 3 hours caused physiological responses comparable to what is seen in trawl-captured Atlantic cod. Together with a previous study [12], this indicates that the additional physiological stress caused by crowding in the cod-end is an important contributor to the often-observed reduction in fillet quality of cod caught by trawl. It is suggested that the accumulation of residual blood in the white muscles of crowded fish may be the result of insufficient emptying of segmental veins due to the static condition of the muscles during crowding. A complete recovery from exhaustive exercise and extreme crowding most likely requires more than 6 hours.

## Supporting information

**S1 Fig. Extreme crowding of Atlantic cod.** Image showing the extreme crowding of cod in the experimental cod-end. The shape of the closed cod-end resembled a sphere with diameter 58 cm.
(TIF)

**S2 Fig. GLM correlation plots.**
(PDF)

**S1 Model. Model parameters and ANOVA output from the generalized linear models.**
(PDF)

## Acknowledgments

We would like to extend our deepest thanks to Kjell Midling, whose ideas motivated this work and who, although no longer with us, continues to inspire by his example and dedication to

improve the Norwegian fisheries. We would also like to thank Tor H Evensen, (Nofima) for skilful technical assistance and Tatiana Ageeva, Sjurdur Joensen and Torbjørn Tobiassen for help with filleting of fish and sensory evaluation of fillets. The valuable help from the technical staff at the Tromsø aquaculture research station is also gratefully acknowledged.

## Author Contributions

**Conceptualization:** Ragnhild Aven Svalheim, Øyvind Aas-Hansen, Anders Karlsson-Drangsholt, Stein Harris Olsen, Helge Kreutzer Johnsen.

**Data curation:** Ragnhild Aven Svalheim, Karsten Heia.

**Formal analysis:** Ragnhild Aven Svalheim, Karsten Heia.

**Funding acquisition:** Øyvind Aas-Hansen, Helge Kreutzer Johnsen.

**Investigation:** Ragnhild Aven Svalheim, Anders Karlsson-Drangsholt, Helge Kreutzer Johnsen.

**Methodology:** Ragnhild Aven Svalheim, Øyvind Aas-Hansen, Karsten Heia, Anders Karlsson-Drangsholt, Helge Kreutzer Johnsen.

**Project administration:** Ragnhild Aven Svalheim, Øyvind Aas-Hansen.

**Resources:** Ragnhild Aven Svalheim.

**Software:** Karsten Heia.

**Supervision:** Øyvind Aas-Hansen, Stein Harris Olsen, Helge Kreutzer Johnsen.

**Validation:** Ragnhild Aven Svalheim, Helge Kreutzer Johnsen.

**Visualization:** Ragnhild Aven Svalheim.

**Writing – original draft:** Ragnhild Aven Svalheim.

**Writing – review & editing:** Ragnhild Aven Svalheim, Øyvind Aas-Hansen, Karsten Heia, Anders Karlsson-Drangsholt, Stein Harris Olsen, Helge Kreutzer Johnsen.

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
