## [Decision Letter · Decision Letter 0]

15 Apr 2020

PONE-D-20-04124

Simulated trawling: Exhaustive swimming followed by extreme crowding as contributing reasons to variable fillet quality in trawl-caught Atlantic cod (Gadus morhua)

PLOS ONE

Dear Dr SVALHEIM,

Thank you for submitting your manuscript to PLOS ONE. After careful consideration, we feel that it has merit but does not fully meet PLOS ONE’s publication criteria as it currently stands. Therefore, we invite you to submit a revised version of the manuscript that addresses the points raised during the review process.

Please pay attention to the list of comments by the reviewer; please check the quality of the figures.

We would appreciate receiving your revised manuscript by May 30 2020 11:59PM. To enhance the reproducibility of your results, we recommend that if applicable you deposit your laboratory protocols in protocols.io, where a protocol can be assigned its own identifier (DOI) such that it can be cited independently in the future. For instructions see: http://journals.plos.org/plosone/s/submission-guidelines#loc-laboratory-protocols

We look forward to receiving your revised manuscript.

Kind regards,

Ruud van den Bos, Ph.D.

Academic Editor

PLOS ONE

Journal Requirements:

2. Please include your tables as part of your main manuscript and remove the individual files. Please note that supplementary tables (should remain/ be uploaded) as separate "supporting information" files

Reviewers' comments:

Reviewer's Responses to Questions

**Comments to the Author**

1. Is the manuscript technically sound, and do the data support the conclusions?

Reviewer #2: Yes

2. Has the statistical analysis been performed appropriately and rigorously? 

Reviewer #2: Yes

3. Have the authors made all data underlying the findings in their manuscript fully available?

Reviewer #2: Yes

4. Is the manuscript presented in an intelligible fashion and written in standard English?

Reviewer #2: Yes

5. Review Comments to the Author

Reviewer #2: The authors try to simulate the different processes that involve a trawl situation and see the cumulative effects of these different processes on metabolic parameters and flesh quality subsequently of wild-caught Atlantic cod. This is achieved by simulation of exhaustive swimming followed by extreme crowding. Secondary was investigated whether post-stress recuperation could mitigate the negative effects on stress and fillet quality.

The manuscript is generally well written and the level of English is adequate.

The use of figures to explain the experimental design is a bonus and makes it much easier to comprehend the experiment.

I agree with a previous reviewer that the word fillet quality envelops more than is analyzed in this paper. Fillet colour and pH might be better to address what has been measured.

Discussion & Conclusion

The discussion and conclusions are short and to the point and don’t seem to contain irrelevant discussion or unsupported conclusions anymore.

Figures:

Unfortunately the quality of the figures is poor. I don’t know if this is due to the uploading process, but the quality needs to be improved. In this state the text in the figures is hardly readable. No significant differences are shown in the figures, I would suggest adding these so the figures show all information without reading the text.

Minor changes:

Line 57: an, a

Line 58: Short-term

Line 67: something is missing in this sentence, referring to “Both” and only one argument is given

Line 92: live, alive

Line: 102: non, of

Line 148: taken out, captured

Line 153: m s-1

Line 188: (4x4 cm), hole size I assume?

Line 189: Mars, March

Line 206: obtained from, measured in

Line 245: assaying, analyses

Line 326: haemosglobin, haemoglobin

Line 331: (N= 21, N = 21

Line 342: (N= 21, N = 21

Line 407: space

6. PLOS authors have the option to publish the peer review history of their article (what does this mean?). If published, this will include your full peer review and any attached files.

Reviewer #2: No

---

## [Author Response · Author response to Decision Letter 0]

12 May 2020

We thank the reviewers for their generous comments on the manuscript and have edited the manuscript to address their concerns. 

Comment from reviewer: I agree with a previous reviewer that the word fillet quality envelops more than is analyzed in this paper. Fillet colour and pH might be better to address what has been measured.

Answer: Thank you for this comment. We have now changed the word “fillet quality” to “fillet colour” where applicable. 

Comment from reviwer: Figures:

Unfortunately the quality of the figures is poor. I don’t know if this is due to the uploading process, but the quality needs to be improved. In this state the text in the figures is hardly readable. No significant differences are shown in the figures, I would suggest adding these so the figures show all information without reading the text.

Answer: Thank you for this comment. I am uncertain why the quality of the figure was so poor the previous version. I have redone all figures, and hopefully they are readable now. Significant differences (based on confidence intervals from the GLM) is now shown in all figures. 

Line 57: an, a

ok

Line 58: Short-term

ok

Line 67: something is missing in this sentence, referring to “Both” and only one argument is given

ok

Line 92: live, alive

ok

Line: 102: non, of

ok

Line 148: taken out, captured

ok

Line 153: m s-1

ok

Line 188: (4x4 cm), hole size I assume?

ok

Line 189: Mars, March

Line 206: obtained from, measured in

ok

Line 245: assaying, analyses

ok

Line 326: haemosglobin, haemoglobin

ok

Line 331: (N= 21, N = 21

ok

Line 342: (N= 21, N = 21

ok

Line 407: space

ok

---

## [Editor Report · Decision Letter 1]

19 May 2020

Simulated trawling: Exhaustive swimming followed by extreme crowding as contributing reasons to variable fillet quality in trawl-caught Atlantic cod (Gadus morhua)

PONE-D-20-04124R1

Dear Dr. SVALHEIM,

We are pleased to inform you that your manuscript has been judged scientifically suitable for publication and will be formally accepted for publication once it complies with all outstanding technical requirements.

With kind regards,

Ruud van den Bos, Ph.D.

Academic Editor

PLOS ONE
---

## [Editor Report · Acceptance letter]

3 Jun 2020

PONE-D-20-04124R1 

Simulated trawling: Exhaustive swimming followed by extreme crowding as contributing reasons to variable fillet quality in trawl-caught Atlantic cod (Gadus morhua) 

Dear Dr. SVALHEIM:

I'm pleased to inform you that your manuscript has been deemed suitable for publication in PLOS ONE. Congratulations! Your manuscript is now with our production department. 

Kind regards, 

on behalf of

Dr. Ruud van den Bos 

Academic Editor

PLOS ONE